# Thymoquinone Protects Neurons in the Cerebellum of Rats through Mitigating Oxidative Stress and Inflammation Following High-Fat Diet Supplementation

**DOI:** 10.3390/biom11020165

**Published:** 2021-01-27

**Authors:** Aziza Alrafiah

**Affiliations:** Department of Medical Laboratory Technology, Faculty of Applied Medical Sciences, King Abdulaziz University, Jeddah 21589, Saudi Arabia; aalrafiah@kau.edu.sa; Tel.: +966-1640-1000 (ext. 23495)

**Keywords:** high-fat diet, inflammatory cytokines, oxidative stress, thymoquinone

## Abstract

High-fat diet (HFD) is a major problem causing neuronal damage. Thymoquinone (TQ) could regulate oxidative stress and the inflammatory process. Hence, the present study elucidated the significant role of TQ on oxidative stress, inflammation, as well as morphological changes in the cerebellum of rats with HFD. Rats were divided into three groups as (1) control, (2) saturated HFD for eight weeks and (3) HFD supplementation (four weeks) followed by TQ 300 mg/kg/day treated (four weeks). After treatment, blood samples were collected to measure oxidative stress markers glutathione (GSH), malondialdehyde (MDA), superoxide dismutase (SOD), and inflammatory cytokines. Furthermore, neuronal morphological changes were also observed in the cerebellum of the rats. HFD rats show higher body weight (286.5 ± 7.4 g) as compared with the control group (224.67 ± 1.78 g). TQ treatment significantly (*p* < 0.05) lowered the body weight (225.83 ± 13.15 g). TQ produced a significant (*p* < 0.05) reduction in cholesterol, triglycerides, high-density lipoprotein (HDL), and low-density lipoprotein (LDL). The antioxidative enzymes significantly reduced in HFD rats (GSH, 1.46 ± 0.36 mol/L and SOD, 99.13 ± 5.41 µmol/mL) as compared with the control group (GSH, 6.25 ± 0.36 mol/L and SOD, 159.67 ± 10.67 µmol/mL). MDA was increased significantly in HFD rats (2.05 ± 0.25 nmol/L) compared to the control group (0.695 ± 0.11 nmol/L). Surprisingly, treatment with TQ could improve the level of GSH, MDA, and SOD. TQ treatment significantly (*p* < 0.05) reduced the inflammatory markers as compared with HFD alone. TQ treatment minimizes neuronal damage as well as reduces inflammation and improves antioxidant enzymes. TQ can be considered as a promising agent in preventing the neuronal morphological changes in the cerebellum of obese populations.

## 1. Introduction

Various chronic diseases, such as diabetes, cardiovascular diseases, and cancer, which plague our current times, are majorly influenced by obesity and overweight [1]. In addition, researchers have noted that the risk of neurodegenerative diseases escalates with obesity [2]. The elevated commonness of neurodegenerative disorders could have a significant influence on the life conditions of the patients. Several studies have started exploring the pathophysiology and prevention of neurodegenerative diseases, even though their mechanisms have not been clarified yet. Moreover, there is a distinct parallel relationship between the developments in neurological disease, such as obesity, which has been indicated by significant research [3]. Afflictions to the brain are linked with metabolic alterations generated by obesity. These afflictions could prompt neuronal death, either by apoptosis or cell necrosis, in addition to changes in the neuron’s synaptic plasticity [4,5]. Obesity now is ranked as the fifth leading cause of death worldwide, and its global spread is alarming [6]. The reason to identify overweight and obesity as liable aspects for the progress of Alzheimer’s disease (AD) and Parkinson’s disease (PD) is the high probability for obese patients to develop type 2 diabetes mellitus (DM2) [7], which, in turn, is linked to neurodegenerative diseases [2]. This fact is also explained by the existing relation amidst obesity and insulin resistance that has a crucial part in the development of dementia [3,8,9]. Inflated levels of pro-inflammatory cytokines elevate inflammation that consecutively induces cognitive deficiency [10,11]. The abundant increase of visceral adipose tissue is affiliated with obesity, specifically in adipocyte size, and this elevation extremely impacts the adipose tissue performance [5,12]. White adipose tissue excretes an assortment of pro-inflammatory and anti-inflammatory factors [12], among them are the adipokines (adiponectin and leptin) and cytokines [5,13,14]. Human metabolic syndrome can be modeled by metabolic disorders and diet-induced obesity in rodents, which can be caused by the consumption of HFD [14,15].

Recently, herbal medicine has been broadly used in the treatment of many neurodegenerative disorders. Some of the many advantages to herbal remedies are that they exert minimal side effects, are widely available and easy to administrate [16,17]. The scientific name of *Nigella sativa* (NS) is an annual plant which is known as black seed or cumin. The plant belongs to Ranunculaceae family [18,19]. NS has been shown to have anti-inflammatory, antioxidative, and neuroprotective effects [18]. Extracts of *NS* have been noted to shield the frontal cortex and brain stem from toluene-induced degeneration in rats [19,20]. Research has exhibited that TQ, the active compound of *NS*, has acetylcholinesterase inhibitory effect and shields cultured rat primary neurons against Aβ-induced neurotoxicity [19]. TQ has significant therapeutic properties, such as reducing inflammatory mediators, and exhibits anti-inflammatory effects on encephalomyelitis [11]. Oxidative stress is triggered by free radicals. This phenomenon is evidently known to be increased by obesity [21]. The induced release of abundant amounts of reactive oxygen species (ROS) and pro-inflammatory cytokines has been implicated throughout chronic activation of the microglial cells, in several neurodegenerative diseases such as Alzheimer’s disease [22,23]. While treatment with TQ demonstrated an immense notable decline in malondialdehyde (MDA) level, this could be due to antioxidative properties [18]. Therefore, we hypothesized that TQ may have a significant impact on neuronal morphological changes in the cerebellum of rats following supplementation with HFD and could be one of the therapeutic approaches to improve their brain functions through its antioxidant and anti-inflammatory activity. 

## 2. Materials and Methods

### 2.1. Animals

Adult Albino Wistar rats (*n* = 15) (6–8 weeks), weighing 160–200 g were acquired from the Animal Experimental Unit of King Fahd Centre for Medical Research (KFMRC), King Abdulaziz University, Jeddah, Saudi Arabia. Polypropylene cages (size: 32 × 24 × 16 cm) were used to keep the rats. Rat bedding with paddy was used and cleaned at three-day intervals. Pellet diet, water, 12 h of light cycle was supplied throughout the experimental period. The rats can access food and water freely in a preserved controlled facility in temperature (24 °C ± 1 °C) with 55% ± 10% humidity. The experiment was designed with correspondence to the codes of the guidelines for Ethical Conduct in the Care and Use of Animals. Experimental conduct and handling were authorized via the Animal Ethics division within the Ethics Committee of Biomedical Research-Faculty of medicine at King Abdul Aziz University, ethical approval number (186-18 [HA-02-J-008]). The experiment was executed in consensus with the guidelines of dealing with experimental animals that are followed in KFMRC, KAU, Jeddah, Saudi Arabia, which are in accordance with the Canadian Council for animal safety and health care. The black cumin seeds were collected from the local retailer (Kingdom of Kif factory for packing foodstuffs, 63298), Jeddah, KSA. TQ was extracted from NS in the Department of Natural Products and Alternative Medicine, Faculty of Pharmacy, KAU, Jeddah. The TQ concentration of NS was analyzed by high-performance liquid chromatography (HPLC).

### 2.2. Experimental Groups

A total of 15 animals were divided into three groups: (1) control (*n* = 5), (2) high-fat diet (HFD) (*n* = 5) (6 mg/day) for eight weeks, and (3) high-fat diet (HFD) (*n* = 5) for four weeks with TQ (300 mg/kg/day) for four weeks. For the induction of obesity in rats, different kinds of diets had been used, control chow diet and HFD: (a) Regular rodent chow diet: The assembled diet was in accordance with [16]. The ratio of compounds in a regulated rat diet is 65% CHO (60% starch and 5% sucrose), 20% protein, 5% fat, 5% minerals and vitamins, and 5% fibers. This diet has a metabolic efficiency of 2813 kcal/kg, with 8% coming out of fat. (b) The saturated HFD: The diet is formed of 20% saturated fat (butter) obtained from local retail in Jeddah, 2% cholesterol and 0.5% bile acid salts. (c) The high saturated fat diet + the TQ Nigella seeds extract: The TQ dose of (300 mg/kg/day) was prepared with distilled water administered by intragastric intubation [20]. 

### 2.3. Measurement of Inflammatory Markers

Rat Cytokine/Chemokine (Cat. No. RECYTMAG-65K, USA) was used to measure the levels of IL-1β, IL-6, TNF-α in serum according to the manufacturer’s protocol. Luminex 200 machine and Milliplex Analyst software were used for data analysis at the Neuroscience Unit in KAUH. 

### 2.4. Measurement of Oxidative Stress Markers

Malondaldehyde (MDA) was determined in serum spectrophotometrically as thio barbituric acid reactive substances (TBARS) [21]. GSH level was determined in serum as described by Ellman [22]. SOD activity was estimated according to the method adopted by Marklund [23].

### 2.5. Morphological Changes 

Following HFD supplementation, rats had undergone cardiovascular perfusion with PBS and paraformaldehyde according to the procedure of [24]. The brain was harvested and used for tissue processing. Following fixation, coronal tissue sections were cut. Then, haematoxylin and eosin staining was performed by immersing the glass slides in xylene and gradient of alcohol in order to deparaffinize and rehydrate the sections. The procedure was done in an automatic tissue stainer. Finally, the slides were covered with a thin glass cover slip using a mounting medium and studied under a light microscope. 

### 2.6. Statistical Analysis

The results were represented as mean ± standard deviation (SD) and were analyzed by the Statistical Package of Social Sciences (SPSS) program version 16. We used one-way analysis of variance (ANOVA). ANOVA was followed by Bonferroni’s multiple comparison test to compare different groups [25]. 

## 3. Results

### 3.1. HFD on Body Weight and Cholesterol Profile 

To evaluate the effect of HFD on the body weight of rats and their serum cholesterol profile. The rats supplemented for eight weeks with HFD showed a significant increase in body weight (*p* < 0.05) but treatment with TQ with HFD supplementation significantly (*p* < 0.05) lowered the body weight (Figure 1A). Before the supplementation of the HFD, there was no significant body weight difference found among all groups. It indicates that the increase of the body weight depends upon the supplementation of HFD. Followed by weighting the body weight, blood collected from rat orbital sinus and carried out serum cholesterol profile. The results in Figure 1B demonstrate the effect of TQ on lipid profile such as cholesterol, triglycerides, HDL, and LDL. The cholesterol level was significantly (*p* < 0.05) elevated following HFD supplementation as compared with the control group, whereas four-week treatment with TQ significantly reduced the level of cholesterol. The cholesterol level was very high as compared with triglycerides, HDL, and LDL in the HFD supplementation group. The level of triglycerides and LDL were significantly (*p* < 0.05) higher in HFD animals, but HDL was increased slightly as compared with those on a normal diet. The body weight and cholesterol were regulated by the TQ treatment and reduced significantly (*p* < 0.05) (Figure 1B). 

### 3.2. HFD on Oxidative Stress Markers 

After investigating the body weight and cholesterol profile, the oxidative stress markers were analyzed. The data show that the glutathione (GSH) (Table 1) level was significantly (*p* < 0.05) reduced following HFD supplementation, whereas the level of GSH significantly increased following treatment with TQ. Interestingly, malondialdehyde (MDA) (Table 1) level was significantly lowered in the HFD supplemented groups as compared with the control group animals. However, after treatment with TQ, the amount of MDA reached the level of the control. HFD supplementation did not have an effect on superoxide dismutase (SOD) (Table 1) levels as in GSH and MDA. HFD results in a significant reduction in SOD as compared with the control group. At the same time, treatment with TQ significantly increased the level of SOD in the treated group (*p* = 0.001). 

### 3.3. HFD on Inflammatory Markers 

After analyzing the antioxidant markers, four inflammatory cytokines such as IL-1β, IL-6, IL-10, and TNF α were analyzed from rat serum. Results show consistent changes across all four markers. 

The data show that IL-1β (Figure 2A) increased significantly (*p* < 0.05) in the HFD but not in the control animals, whereas TQ treated groups showed significantly reduced IL-1β level. In the HFD supplemental groups, the IL6 (Figure 2B) was extremely higher compared to the control, but was reduced following TQ treatment. IL-10 (Figure 2C) and TNF-α (Figure 2D) markers also increased significantly in the HFD group but significantly reduced in the TQ treatment group (*p* < 0.05). The four markers confirmed the inflammation after supplementing with HFD.

### 3.4. HFD on Neuronal Morphology in the Cerebellum 

The neuronal morphological changes were performed using H&E staining after biochemical analysis. The granular layers and Purkinje cells were intact, i.e., the round shape of the cell membrane, the visible nucleus and axons in the control cells (Figure 3A). The group supplementation with HFD shows non-visible cell membrane and nucleus irregular shapes and restructure of axons (Figure 3B). However, after treatment with TQ, neuronal damage in granular and Purkinje cells in the cerebellum were minimized (Figure 3C). 

## 4. Discussion

In the present study, we demonstrate that a high-fat diet causes obesity in terms of increasing body weight as well as cholesterol levels, leading to oxidative stress by minimizing the antioxidant enzyme levels and elevating the inflammatory cytokine levels, which ultimately provokes neuronal damage in the cerebellum of rats. However, TQ has been able to reduce body weight, cholesterol levels, improve antioxidant enzymes, and mitigate inflammatory cytokines along with cerebellum neuroprotection. Based on previous studies, it was stated that a high intake of HFD significantly causes obesity. Obesity is one of the risk factors for cognitive functions in terms of learning and memory deficits and is associated with symptoms of AD where HFD potentiates the onset of microglial activation, the main source of neuroinflammation [26]. A study by [27] revealed that hypercholesterolemia might be a high risk factor for developing AD. In line with these results, another study confirmed that HFD affects memory and causes tau pathology in female offspring during gestation and lactation [28]. Mostly, clinical complications of obesity lead to learning and memory deficits in obese patients [29]. Note that HFD plays a major role in controlling brain functions and behaviors in both humans and animals. It is, therefore, necessary to find a valuable therapeutic approach to reduce this burden by using herbal products. It was, therefore, necessary to study the role of TQ in rat cerebellum on body weight, cholesterol profile, antioxidant enzymes, anti-inflammation, and morphological changes. Interestingly, the results of this study show a significant correlation between total body weight gain and average daily intake. Rats fed with HFD increased cholesterol levels for eight weeks, resulting in higher serum TC, HDL, and LDL levels. It is suggested that there is a correlation between weight gain and cholesterol profile. The reason for increasing the cholesterol level in the circulating blood is that the body cannot metabolize lipids after HFD supplementation [30]. In agreement with our current results, Ref. [31] reported a significant increase in total triglyceride levels following HFD intake after two weeks. Surprisingly, Ref. [32] reported that HFD-fed male mice respond faster to high-fat diet and gain weight than females. It has been stated that long-term food intake could increase body weight [33,34]. In the present study, treatment with TQ significantly reduced body weight along with decreased serum cholesterol profile such as TC, HDL, and LDL. In agreement with previous studies, TQ treatment could decrease serum levels of LDL, TC, and HDL [35]. The results show consistent changes across all four markers [36]. It is interesting that TQ plays a major role in regulating body weight and cholesterol. In our perspective, high cholesterol levels can trigger oxidative stress and then lead to a reduction in antioxidant enzymes. We also conducted analyzes of antioxidant enzymes such as GSH and SOD in the present study. 

In fact, hypercholesterolemia may lead to increased reactive oxygen species (ROS), which may cause cytotoxic effect by reducing the amount of antioxidative enzymes [37]. The present results suggest that HFD rats show a significant reduction in GSH and SOD. In the research line, defense mechanisms of the glutathione pathway significantly altered in HFD as reduced in GSH and SOD. In the line of research, defense mechanisms of glutathione pathway significantly altered in HFD as reduced in GSH and SOD [38]. It was also suggested that early upregulation of genes involved in the production of ROS in adipose tissue causes insufficient production of GPx in the liver [39]. The reduction of the antioxidant enzymes may lead to microglia activation or neuronal damage in the central nervous system. Treatment with TQ for four weeks could increase the level of GSH and SOD as compared with HFD supplementation group. It has been reported that TQ possess strong antioxidant properties and scavenging free-radical production [40]. Based on previous studies, high cholesterol may induce inflammation cytokines in blood serum after intake of HFD [40]. We have also studied in our experiments serum inflammatory markers (IL-1β, IL-6, IL-10, and TNF-α). 

Consumption of high-fat diets and calories seems to be a key factor in obesity [41]. However, the mechanisms underlying the triggering of obesity inflammation have not been elucidated. Reference [25] reported that HFD-induced obesity is closely related to chronic inflammation characterized by activation of inflammatory signaling pathways. These inflammatory signaling pathways are activated by the innate immune response of serum cholesterol and fatty acids [42]. The results of the inflammatory markers show a significant increase in the HFD supplementation group. The levels of inflammatory cytokines were elevated in HFD animals. These data are supported by [43]. His study found that the levels of IL-1β and IL-6 increased. Consistent evidence reported by [44] shows that the oral administration of TQ significantly reduces the levels of different inflammatory mediators. It could be suggested that one of the crucial factors to trigger inflammation is serum cholesterol and oxidative stress. 

Numerous reports suggest the anti-inflammatory activity of TQ [45,46]. TQ mediates anti-inflammatory activity through heme oxygenase-1 (HO-1), inhibits toll-like receptor 4 and reduces the level of pro-inflammatory cytokines. It displays anti-inflammatory activity by inhibiting phosphatidylinositol 3-kinase phosphorylation and increased AMPK [47]. The possible inflammation factor is cyclooxygenase [47]. The process of the inflammatory response is considered as a protective role in the biological processes, which are regulated by endogenous mediators to eradicate harmful stimuli. Cytokines and chemokines are the most common inflammatory mediators in physiological functions. Macrophages and neutrophils are major cells that produce inflammatory cytokines in injured tissue [48]. In addition, prostaglandins (PGs) biosynthesis and cyclooxygenase (COX) enzyme are critically organized in inflammatory processes [49]. In the present study, inflammatory cytokines increased after supplementation with HFD but reduced the level of cytokines after treatment with TQ. It could be possible for TQ to regulate cyclooxygenase activity. The study of [47] reported that TQ inhibits COX-2 and NF-kB induction. The previous study supported TQ’s ability to suppress COX-2 and reduce inflammation [50]. In fact, nitric oxide (NO) generated by inducible nitric oxide synthase (iNOS) plays a major role in the inflammation process [51]. NO also considers free radicals that cause tissue damage through oxidative processes [52]. TQ is believed to be able to influence NO to reduce inflammatory cytokines. Further study supported TQ inhibiting NF-kappaB and NO activation [44]. Consistent with this finding, [53] reported that TQ inhibited NO production by reducing the expression of iNOS to reveal its anti-inflammatory response. It has been suggested that by inhibiting COX, NO, and NF-kappaB, TQ has strong anti-inflammatory potential. While TQ mitigates cholesterol levels, oxidative stress, and inflammatory cytokines, we are very interested in investigating neuronal morphological changes in the cerebellum. 

The morphological changes were studied in the cerebellum, followed by the biochemical assay. Eight weeks of HFD rat supplement show neuronal damage in the granular layers and Purkinje cells, while TQ-treated neuronal damage in the cerebellum may be reduced. Consistent evidence suggests that patients with clinical obesity showed reduction and enlargement of the frontal lobe in both gray and white matters [54]. Increased body mass index is also associated with decreased brain volume [55]. The mechanisms underlying neuronal damage mediation are considered oxidative damage and inflammation [56]. Oxidative damage is defined as damage that occurs during oxidative stress on biomolecules. Oxidative stress is a process in which the imbalance between ROS production and reactive nitrogen species and antioxidant enzymes and their contribution to pathogenesis has been widely studied [57]. The mechanisms underlying oxidative damage in obesity brain are still poorly understood, although many experiments have been reported in humans and animals [43,58]. Based on the previous studies [59], TQ may reduce neuronal damage by inhibiting kinase-regulating apoptosis signal1 (ASK1) that triggered pathways of JNK and p38 MAPK. Oxidative stress and inflammatory cytokines like TNF-α are regulated by these pathways. To understand how TQ prevents neuronal damage, these pathways need to be studied in the future. 

Under ROS overproduction, possibly due to obesity-induced inflammation, someone would expect an increase of antioxidant enzymes to counteract ROS overproduction. This did not happen in HFD rats, but TQ reversed this phenomenon. It can be argued that the body mass influences ROS production. The decreased body weight of the TQ treated group could, therefore, be associated with a decreased level of oxidative stress markers. This is consistent with what was discovered by Roberta et al. (2017) that changes in levels of oxidative stress markers analyzed after bariatric surgery are correlated with body mass, affecting reactive oxygen species production [60].

## 5. Conclusions

Thymoquinoneon controls the body weight, decrease the levels of the cholesterol profile, increase the level of antioxidant enzymes, mitigate inflammatory cytokines, and prevent neuronal damage following supplementation with high-fat diet. We could suggest that TQ could be one of the therapeutic approaches to improve brain functions and behaviors in obese or overweight people based on the present findings. Further studies are needed to be done in the future to evaluate the effect of the daily clinical use of TQ. 

## Figures and Tables

**Figure 1 biomolecules-11-00165-f001:**
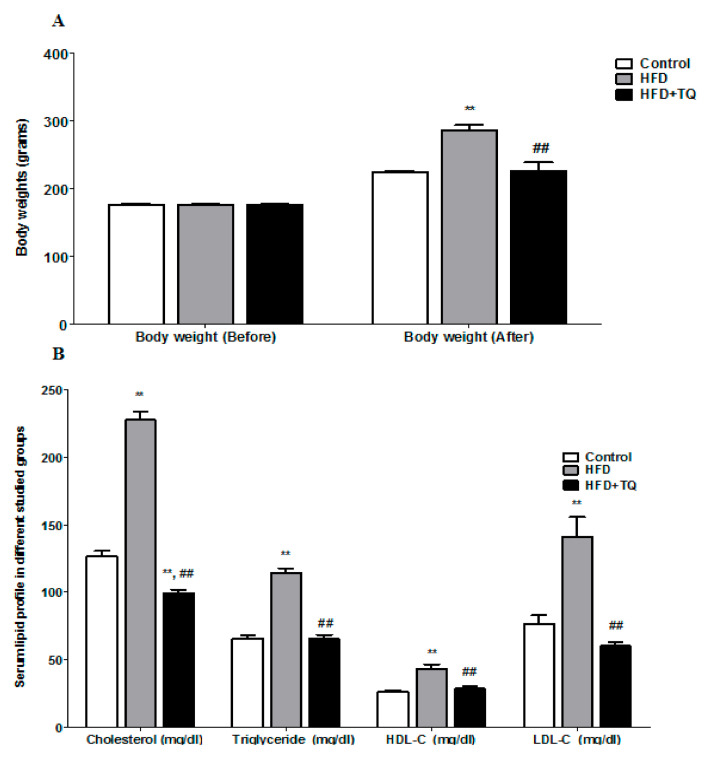
(**A**) Comparison of the body weight before and after treatment in different studied groups. (**B**) Comparison of the lipid profiles in different studied groups. Data are expressed as ± SEM. **: significance versus control (*p* = 0.001); ##: Significance versus HFD (*p* = 0.001).

**Figure 2 biomolecules-11-00165-f002:**
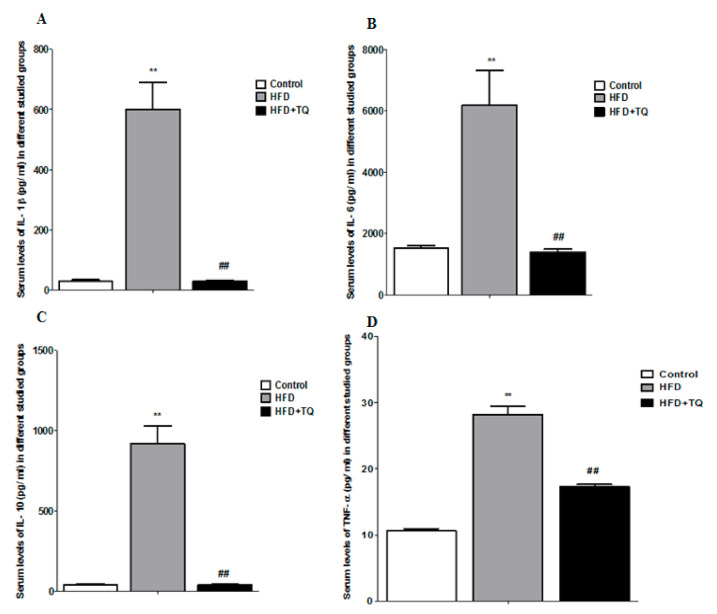
The inflammatory cytokines in different studied groups. (**A**) Comparison of serum levels of IL-1β (pg/mL) in different studied groups. (**B**) Comparison of serum levels of IL-6 (pg/mL) in different studied groups. (**C**) Comparison of serum levels of IL-10 (pg/mL) in different studied groups. (**D**) Comparison of serum levels of TNF-α (pg/mL) in different studied groups. Data are expressed as ± SEM. **: significance versus control (*p* = 0.001); ##: Significance versus HFD (*p* = 0.001).

**Figure 3 biomolecules-11-00165-f003:**
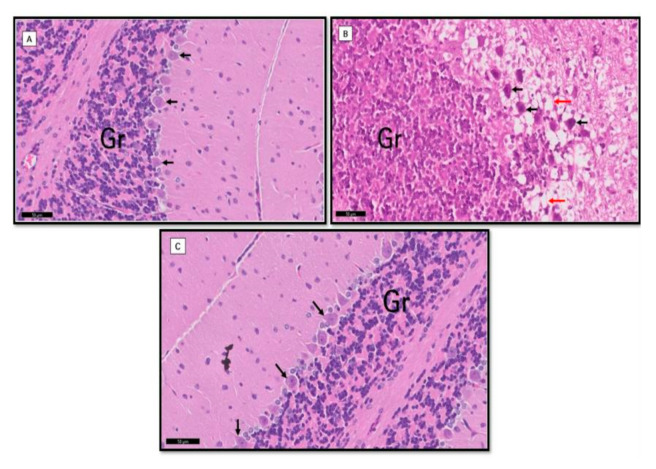
Comparison of morphological changes in the cerebellum in the different studied groups. Group I (**A**): The Purkinje with apical dendrites (black ↑) in one row, pale stained nuclei, and prominent nucleoli. Notice the tightly packed granule cells (Gr) and cerebellar islands between them. Group II (**B**): Purkinje cells appeared with deep acidophilic cytoplasm and deeply stained pyknotic nuclei (↑). Notice vacuolation within the nearby molecular layer (red ↑). Granular layers (Gr) showed marked spaces between the granular cells. Group III (**C**): Apparently normal Purkinje cells (black ↑) and granular cells (Gr) nearly to the control.

**Table 1 biomolecules-11-00165-t001:** Comparison of glutathione (GSH), superoxide dismutase (SOD), and malondialdehyde (MDA) serum levels in the different studied groups.

	Control	HFD	HFD + TQ
**GSH (mol/L)**	6.25 ± 0.36	1.46 ± 0.36 *	10.06 ± 1.09 #
**SOD (u/mL)**	159.67 ± 10.67	99.13 ± 5.41 *	177.67 ± 15.02 #
**MDA (nmol/L)**	0.695 ± 0.11	2.05 ± 0.25 *	0.690 ± 0.15 #

Values are represented as mean ± SD, *n* = 6. ∗ compared to control group (*p* < 0.05). # compared to HFD group (*p* < 0.05).

## Data Availability

Data is contained within the article.

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
