# Peer review of "Thymoquinone Protects Neurons in the Cerebellum of Rats through Mitigating Oxidative Stress and Inflammation Following High-Fat Diet Supplementation"

_biomolecules, 2021, doi:10.3390/biom11020165_

Round 1

Reviewer 1 Report

Dear Authors, 

Thank you very much for this resubmission. However, in a previous revision, I asked you a lot of questions regarding the methods but you haven’t replied to these questions and modified on the manuscript. 

Please, resubmit the manuscript highlighting the changes and include your answers to my comments (statistical analyses, free radical assessment,...). 

Thank you very much. 

Author Response

Dear reviewer,

Thank you for allowing me to submit a revised draft of my manuscript titled [Thymoquinone Protects Neurons in the Cerebellum of Rats through Mitigating Oxidative Stress and Inflammation Following High Fat Diet Supplementation] to [biomolecules]. I appreciate the time and effort that you have dedicated to providing your valuable feedback on my manuscript. I am grateful to the reviewers for their insightful comments on my paper. I have been able to incorporate changes to reflect most of the suggestions provided by the reviewers. I have highlighted the changes within the manuscript.

Here is a point-by-point response to the reviewers’ comments.

Comment 1: Please, resubmit the manuscript highlighting the changes and include your answers to my comments (statistical analyses, free radical assessment,...).

Response: Thank you for pointing this out. I included the numerical data related to the free radical assessment. It is highlighted in the abstract. Moreover, I added a table (page 4, highlighted) showing numerical data for TQ's effect on the oxidative stress markers (GSH, SOD, and MDA).

Also, I have improved the abstract, added one paragraph to explain the hypothesis. Moreover, two sections were added to the method for measuring oxidative stress markers and statistical analysis. Finally, the conclusion was improved as well. All the changes are highlighted in the revised manuscript.

Once again, thank you for the time you put in reviewing my paper, and I look forward to meeting your expectations. Since your inputs have been precious, we would like to acknowledge your contribution explicitly in the eventuality of a publication.

Sincerely,

Aziza

Reviewer 2 Report

The article is rather weak, the results are not very novel and predictable.

Author Response

Dear reviewer,

Thank you for allowing me to submit a revised draft of my manuscript titled [Thymoquinone Protects Neurons in the Cerebellum of Rats through Mitigating Oxidative Stress and Inflammation Following High Fat Diet Supplementation] to [biomolecules]. I appreciate the time and effort that you have dedicated to providing your valuable feedback on my manuscript. I am grateful to the reviewers for their insightful comments on my paper. I have been able to incorporate changes to reflect most of the suggestions provided by the reviewers. I have highlighted the changes within the manuscript.

Here is a point-by-point response to the reviewers’ comments.

Comment 1: [The article is rather weak, the results are not very novel and predictable.]

Response: Thank you for pointing this out. As suggested by the reviewer, I included the numerical data related to the paper results in the results section of the abstract. It is highlighted in the abstract. Moreover, I added a table (page 4, highlighted) showing numerical data for TQ's effect on the oxidative stress markers (GSH, SOD, and MDA).

Also, I have improved the abstract, added one paragraph to explain the hypothesis. Moreover, two sections were added to the method for measuring oxidative stress markers and statistical analysis. Finally, the conclusion was improved as well. All the changes are highlighted in the revised manuscript.

Once again, thank you for the time you put in reviewing my paper, and I look forward to meeting your expectations. Since your inputs have been precious, we would like to acknowledge your contribution explicitly in the eventuality of a publication.

Sincerely,

Aziza

Reviewer 3 Report

To:

Editorial Board

Biomolecules

Title: “Thymoquinoneon Protects Neurons in the Cerebellum of Rats through Mitigates Oxidative Stress and Inflammation Following High Fat Diet Supplementation”

Dear Editor,

I read this paper and I think that:

  • The English of the paper should be revised in order to improve the readability of the text. Please revise.
  • The results section of the abstract is really poor. The authors should include the numerical data related to the results of the paper.
  • The role of TQ in daily clinical practice is questionable. I think that the authors should have considered a further group with well-established compounds able to influence lipids metabolism such as statins. This would improve the strength of the results of the paper and improve the novelty of it.
  • The role of nutraceuticals in lipid metabolism control has been well established (see also the paper from Scicchitano P et al. Journal of Functional Foods 2014;6:11-32). The novelty of the paper should be better outlined.
  • The application of the use of TQ in daily clinical practice should be discussed.

Author Response

Dear reviewer,

Thank you for allowing me to submit a revised draft of my manuscript titled [Thymoquinone Protects Neurons in the Cerebellum of Rats through Mitigating Oxidative Stress and Inflammation Following High Fat Diet Supplementation] to [biomolecules]. I appreciate the time and effort that you have dedicated to providing your valuable feedback on my manuscript. I am grateful to the reviewers for their insightful comments on my paper. I have been able to incorporate changes to reflect most of the suggestions provided by the reviewers. I have highlighted the changes within the manuscript.

Regarding the English, I have sent it to scribendi website for proofreading. 

Here is a point-by-point response to the reviewers’ comments attached.  

Round 2

Reviewer 1 Report

Dear Authors,

Thank you very much for resubmitting the article. It has improved.

Thank you.

Reviewer 2 Report

Author of the article "Thymoquinoneon protects neurons in cerebellum of rats through mitigates oxidative stress and inflammation following high fat diet " presented an edited version, which additionally describes methods for measuring markers of oxidative stress and statistical analysis. A table of changes in GSH, SOD and MDA levels is provided. The thymoquinoneon function is more clearly indicated, which is to improve brain functions and behaviors in obese or overweight people. We can agree with this on the basis of the presented experimental data. In this regard, the article is recommended for publication

Reviewer 3 Report

THe authors addressed my previous comments. The paper improved.

This manuscript is a resubmission of an earlier submission. The following is a list of the peer review reports and author responses from that submission.

Round 1

Reviewer 1 Report

Dear Authors,

This is a good article analyzing the effects of thymoquinone in a high-fat diet. The methods are appropriate, and the conclusions are supported by the results. I have some comments:

Material and Methods. Please, include a section related to Statistical Analysis. Which program have you used? Which methods?
Results. Figure 1B, LDL the TQ+HFD group is significant respect to HFD but not Control. Do you have any explanation? On the graph there is a clear difference. The same with the levels of TNF-alpha.

Results. Have you checked oxidative stress levels? Based on the results with antioxidants enzymes. It will be interesting to check if, effectively, TQ reduces free radicals levels.

Thank you very much.

Reviewer 2 Report

REVIEW

of the article of Aziza Alrafian “Thymoquiinoneon protects Neurons in the Cerebellum of Rats through Mitigates Oxidative Stress and Inflammation Following High Fat   Supplementation”

Author of reviewed paper explores the effect of the natural component “Thymoquiinoneon (TQ), derived from Nigella sativa, which is an annual herbaceous plant cultivated in South West Asia.

  1. sativa seed extract, fixed oil and essential oil showed a wide spectrum of favorable biological activities, the most prominent being antioxidant, anti-inflammatory, antibacterial, hepatoprotective, antimutagenic and antitumor activities.

In the presented study, the e levels of antioxidant enzymes and inflammatory cytokines such as IL-1b, I L-6 and TNFauthor used this extract to prevent obesity in rats fed a super-caloric diet. Based on the well known antioxidant and inflammatory properties of this drug, the author investigated the change in serum when animals fed by high fat diet and Thymoquiinoneon. Cholesterol, HDL and LDL levels were measured in parallel. Morphological changes in the cerebellum were carried out as well. However, the research would be more valuable if the author also measured markers of the antioxidant and pro-inflammatory properties of TQ specifically in cerebellum. In addition, it would be extremely useful to study these properties of TQ in animals that are already obese.

The article presents quite expected results. But despite this, it contains quite serious shortcomings.

  1. First of all, description of the methods used by the author are very superficially presented. A detailed description of the antioxidant enzyme and cytokine assays should be given.
  2. The lack of this information in the method section does not allow us to assess where these parameters are measured, in the structure of the brain or in plasma. This is only clarified in the results section.
  3. Figure 2 requires full correction. Figure 2C is missing, the ordinate of Fig 2B is erroneously represented by SOD instead of MDA.
  4. The Discussion should be edited. It contains many repetitions and wears quite chaotic nature.
  5. Section titles 3.1. 3.2, 3.3 and 3.4 also need to be edited.
  6. English must be edited by a professional editor.

As presented, the article cannot be published and must be seriously edited.

Reviewer 3 Report

To:

Editorial Board

Biomolecules

Title: “Thymoquinoneon Protects Neurons in the Cerebellum of Rats through  Mitigates Oxidative”

Dear Editor,

I read this paper and I think that:

  • The paper is misleading: the author declared to evaluate the role of TQ on cerebellum. Indeed, main results are concentrated on oxidative stress which did not exclusively impact on cerebellum but rather on the whole body. The author did not attempt any objective comparative evaluation of the histological alterations in the cerebellum nor they performed evaluation of the correlation between histological alterations and oxidative stress biomarkers.
  • The English of the paper should be revised due to typos. A native English speaker should be considered.
  • The results section of the abstract should be implemented: please include more numerical data. Do not leave p-values alone.
  • Acronyms should be expressed at their first mention both in the abstract and in the main text. Please revise the paper.
  • The role of nutraceuticals should be discussed in clinical conditions. Please discuss such a point in a dedicated limitation section.
  • Page 4 lines 123-124: reporting the aims of the paper in the results section is redundant. Please delete this sentence.